# Functional Gait Recovery after a Combination of Conventional Therapy and Overground Robot-Assisted Gait Training Is Not Associated with Significant Changes in Muscle Activation Pattern: An EMG Preliminary Study on Subjects Subacute Post Stroke

**DOI:** 10.3390/brainsci11040448

**Published:** 2021-04-01

**Authors:** Francesco Infarinato, Paola Romano, Michela Goffredo, Marco Ottaviani, Daniele Galafate, Annalisa Gison, Simone Petruccelli, Sanaz Pournajaf, Marco Franceschini

**Affiliations:** 1Neurorehabilitation Research Laboratory, IRCCS San Raffaele Roma, 00163 Rome, Italy; francesco.infarinato@sanraffaele.it (F.I.); paola.romano@sanraffaele.it (P.R.); marco.ottaviani@sanraffaele.it (M.O.); daniele.galafate@sanraffaele.it (D.G.); annalisa.gison@sanraffaele.it (A.G.); simone.petruccelli@sanraffaele.it (S.P.); sanaz.pournajaf@sanraffaele.it (S.P.); marco.franceschini@sanraffaele.it (M.F.); 2Department of Human Sciences and Promotion of the Quality of Life, San Raffaele University, 00166 Rome, Italy

**Keywords:** stroke, robot-assisted gait training, exoskeleton device, neurologic gait disorders, rehabilitation, EMG

## Abstract

Background: Overground Robot-Assisted Gait Training (o-RAGT) appears to be a promising stroke rehabilitation in terms of clinical outcomes. The literature on surface ElectroMyoGraphy (sEMG) assessment in o-RAGT is limited. This paper aimed to assess muscle activation patterns with sEMG in subjects subacute post stroke after training with o-RAGT and conventional therapy. Methods: An observational preliminary study was carried out with subjects subacute post stroke who received 15 sessions of o-RAGT (5 sessions/week; 60 min) in combination with conventional therapy. The subjects were assessed with both clinical and instrumental evaluations. Gait kinematics and sEMG data were acquired before (T1) and after (T2) the period of treatment (during ecological gait), and during the first session of o-RAGT (o-RAGT_1_). An eight-channel wireless sEMG device acquired in sEMG signals. Significant differences in sEMG outcomes were found in the BS of TA between T1 and T2. There were no other significant correlations between the sEMG outcomes and the clinical results between T1 and T2. Conclusions: There were significant functional gains in gait after complex intensive clinical rehabilitation with o-RAGT and conventional therapy. In addition, there was a significant increase in bilateral symmetry of the Tibialis Anterior muscles. At this stage of the signals from the tibialis anterior (TA), gastrocnemius medialis (GM), rectus femoris (RF), and biceps femoris caput longus (BF) muscles of each lower extremity. sEMG data processing extracted the Bilateral Symmetry (BS), the Co-Contraction (CC), and the Root Mean Square (RMS) coefficients. Results: Eight of 22 subjects in the subacute stage post stroke agreed to participate in this sEMG study. This subsample demonstrated a significant improvement in the motricity index of the affected lower limb and functional ambulation. The heterogeneity of the subjects’ characteristics and the small number of subjects was associated with high variability research, functional gait recovery was associated with minimal change in muscle activation patterns.

## 1. Introduction

Wearable Powered Exoskeletons (WPEs) are robotic devices, consisting of sensors, actuators, and control elements, that assist people in performing overground gait with a pre-programmed limbs’ trajectories where the initiation of every single step requires the subject’s active participation [1,2]. The voluntary interaction between the subject and the WPE promotes balance, trunk control, weight transfer, and the loading and unloading of limbs synchronization [3,4,5]. Since the WPEs allow overground gait even in subjects who are not able to keep an upright posture, they can be employed as cutting-edge treatments in neurorehabilitation [6,7,8,9].

Gait rehabilitation based on a WPE referred to as overground Robot-Assisted Gait Training (o-RAGT), is an intensive and repetitive treatment for re-learning gait abilities [10]. The patient is directly involved in the execution of the gait task during the o-RAGT while experiencing a near-normal proprioceptive input and exploring the environment. Existing research on o-RAGT in subjects post stroke is limited, with the patient’s status assessed using a traditional approach based on clinical ordinal rating scales [8]. Although the clinical effects of o-RAGT in neurological subjects are encouraging [7,8,11,12,13,14,15,16,17,18,19], the instrumental assessment of gait biomechanics and muscle activations is a necessity to further understand the effects of this rehabilitation approach [20,21,22,23].

The literature on gait kinematics and surface ElectroMyoGraphy (sEMG) in subjects who conducted the robot-assisted gait training is not homogeneous [24,25,26,27,28,29,30,31,32,33,34,35,36,37]. Most of the studies have involved healthy subjects with the aim of comparing gait assisted by static [24,25,26] or overground [28,29,30] robots to the ecological overground gait. The published results are controversial and highlight how sEMG data are highly dependent on the walking speed, the level and typology of robotic assistance, and thus on the characteristics of the device. Few recent studies have included neurological subjects who underwent gait training with either a static [31,33] or an overground [34,35,36,37] exoskeleton. The gait assisted by a static exoskeleton seems to be associated with a general lowering of muscle activity compared to treadmill walking [31]. These muscular effects depend on the amount of body-weight support, guidance force, and speed [33]. The sEMG analysis during gait assisted by an overground exoskeleton reported controversial results in subjects with spinal cord injury [36] and stroke [34]. Specifically, the study by Androwis et al. [34] analyzed five subjects subacute post stroke (acute event onset time 34.8 ± 34 days) who were able to walk for 10m with the physical therapist’s assistance. The results showed that the WPE allowed patients to preserve their volitional neuromuscular activity and promoted activations of the proximal muscles on the affected side. To the best of our knowledge, only two studies [35,37] applied sEMG before and after a period of o-RAGT treatment. These studies showed that the o-RAGT positively affected gait by increasing the muscle activity [35] and the neuromuscular coordination of lateral symmetry during walking [37]. The literature suggests that sEMG should be extensively used in clinical settings for assessing muscle activity profiles and inter-limb coordination as well as quantifying dynamic motor control parameters in gait [6,38,39,40,41,42]. Moreover, the sEMG analysis has a potential diagnostic capacity and achieves objective information for rational planning and monitoring of the effects of physical therapy [43,44,45]. In particular, hemiplegia, especially in the acute stage, is typically characterized by asymmetrical deficits, muscle weakness, difficulty in producing volitional movements, anomalous muscle activation timing during gait. For these reasons, sEMG techniques are needed to: quantify the intensity of muscle activations; compare activation timings for a single muscle; and assess gait symmetry. However, to date, the translational use of sEMG in rehabilitation centers remains limited [40,42,46,47]. Further studies are needed for assessing benefits and limitations in employing sEMG analysis to patients undergoing o-RAGT to enhance the understanding of the effects of o-RAGT in terms of muscle activity.

This study aimed to evaluate the applicability of a multimodal approach based on both clinical and sEMG assessments for studying the functional gait recovery and the muscular effects of a combination of o-RAGT and conventional therapy in subjects sub-acute post stroke. Specifically, we hypothesized that the gait rehabilitation based on a WPE and conventional therapy would elicit volitional neuromuscular activations and positively change the muscle activation pattern in terms of muscle activity, symmetry and co-activation.

## 2. Materials and Methods

An observational preliminary study was conducted on subjects subacute post stroke who underwent o-RAGT in addition to conventional therapy. This is a secondary analysis from a large multicenter clinical trial assessing the effects of o-RAGT on clinical and biomechanical outcomes [14,15,16] and the barriers to the implementation of a sEMG-based assessment protocol in a clinical context [47]. The data covered by this paper were acquired and processed by a single center (IRCCS San Raffaele Roma, Rome, Italy).

### 2.1. Participants

Subjects were recruited if they met the following inclusion criteria: cerebral stroke (diagnosis confirmed by computed tomography scan and/or magnetic resonance imaging); acute event onset time ≤ 6 months (subacute phase); age between 18–80 years; ability to fit into the WPE (height between 150–190 cm; weight < 100 kg); ability to maintain an upright standing position for at least 60 s; ability to give written consent; compliance with the study procedures. Stroke subjects with the following criteria were excluded from the study: persistent joints contractures that could limit the range of motion during o-RAGT; medical issues (e.g., orthopedic injuries, pain, severe osteoporosis, or severe spasticity); a history of significant problems with skin breakdown or current skin breakdown; cognitive and/or communicative disability; pregnancy; untreated deep vein thrombosis.

Further details on the study methodology, including the enrolment procedures and the setting, were provided in previous papers of the authors [14,15,16,47]. Ethical approval of the treatment and of the evaluation protocol was granted by the Ethics Committee (date: 18 November 2015; code number: 09/15). The study protocol has been registered on clinicaltrials.gov (accessed on 1 April 2021) by the unique identifier number: NCT03395717, and all subjects given informed written consent in accordance with the Declaration of Helsinki.

### 2.2. The Overground Robot-Assisted Gait Training

The subjects conducted 15 sessions of o-RAGT with a WPE (Ekso™, Ekso Bionics, Richmond, CA, USA), 5 times a week. Each session lasted 60 min, including the device donning (approximately 45 min of effective gait training). The Ekso is a wearable bionic suit enabling individuals to stand up and walk over a flat hard surface. By force and motion sensors, the computer interface controls the subject’s movement and translates it into action. According to published clinical studies on the same device [14,15,16], the following WPE settings were used: ProStep Plus™ (each step was triggered by the subject’s transfer load from one leg to the other) and Bilateral Max Assist (the amount of power contribution to legs during walking was totally provided by the robot). Detailed and relevant information on WPE settings is available in the previous paper of the authors [14].

In addition, all subjects conducted daily conventional therapy consisting of physical therapy (e.g., upper limb rehabilitation, functional task practice, muscle strengthening), speech therapy, and occupational therapy.

### 2.3. Outcome Measurements

The subjects were assessed with a multimodal approach based on both clinical and instrumental assessments.

The clinical assessment was performed by administering the following outcome measures before (T1) and after (T2) the period of treatment: Modified Ashworth Scale of the Affected lower Limb (MAS-AL) to evaluate muscle spasticity [48]; Motricity Index of the Affected lower Limb (MI-AL) to measure limb muscle strength [49]; Functional Ambulation Classification (FAC) to evaluate general motor skills necessary for functional ambulation [50]; Trunk Control Test (TCT) to assess the trunk control [51]; the 10-Meter Walking Test (10MWT) to evaluate the maximum walking speed over a short distance [52].

The instrumental assessment was performed for assessing kinematics and electromyographic activity at T1 and T2, during ecological overground gait at a self-selected speed along a 10-m walkway with assistance (e.g., crutches with or without antebrachial support, walker, tripod stick, ankle support orthoses, etc.), and during the first session of o-RAGT (o-RAGT_1_). The o-RAGT_1_ assessment occurred on the same day as the baseline evaluation (T1). Two gait trials were collected for each walking condition. The total duration of the assessment procedure was about 2 h.

An eight-channel wireless sEMG device (FREEEMG 1000-BTS Bioengineering, Milan, Italy) acquired (sampled at 1 kHz, filtered at 8–500 Hz) the activity of the lower limbs’ agonist/antagonist muscles (distal and proximal compartment). After the skin’s abrasion and cleaning with alcohol, the electrodes were placed on Tibialis Anterior (TA), Gastrocnemius Medialis (GM), Rectus Femoris (RF), and Biceps Femoris caput longus (BF) muscles of each leg, according to the SENIAM guidelines [53]. The kinematics was measured using two electrogoniometers for knee joint measurements (BTS Bioengineering, Milan, Italy) and an inertial measurement unit (G-Sensor-BTS Bioengineering, Milan, Italy), placed on the spinous process of the fifth lumbar vertebra, for identifying the gait phases.

The SMART Analyzer software (BTS Bioengineering, Milan, Italy) was employed for signal synchronization, pre-processing and data export. Specifically, data from the inertial sensor were analyzed in comparison with electrogoniometers with the SMART Analyzer software, and the heel strike and toe-off gait cycle events were identified for all gait trials. These temporal events were used for all subsequent sEMG analyses (MATLAB R2019a, The MathWorks Inc., Natick, MA, USA).

### 2.4. Data Processing and Statistical Analysis

The sEMG data were high-pass filtered (20 Hz, 6th-order Butterworth filter, bidirectional) and full-wave rectified [54]. For standardization, the sEMG data were normalized to 100% of a gait cycle based on the temporal events (heel strikes and toe-offs) extracted from the kinematic data. The sEMG signals of each gait cycle have been processed as follows: (1) the sEMG envelope was obtained using a moving average filter (span equal to 120 ms) and normalized at the maximum sEMG amplitude level; (2) the activation threshold identifying onset and offset status of muscle activity was detected as the 20% of minimum–maximum amplitude level distance [55] when kept for at least 50 ms.

A set of sEMG parameters which allow describing the muscle activation pattern during gait was calculated. The literature on sEMG in stroke subjects showed that the muscular patterns of lower limbs are characterized by compensatory mechanisms due to gait impairments [38]. Therefore, the gait tends to be asymmetrical, often requiring excessive muscle co-contraction, and atypical muscular activity. Subsequently, sEMG outcomes to characterizes the gait were extracted: Bilateral Symmetry (BS) [56], the Co-Contraction (CC) [57], and the muscular activity (Root Mean Square, RMS) [58] were calculated.

The BS coefficient for each couple of homologous muscles is an assessment of similarity in muscle behavior between the affected and unaffected sides. It was calculated as the normalized cross-correlation function with a zero-time lag between stride envelopes [56], by applying the following formula to two x and y finite and real series:(1)$${\;\hat{R}}_{\mathrm{xy}}\left(m\right)=\frac{R_{\mathrm{xy}\;\left(m\right)}}{\sqrt{R_{\mathrm{xx}}\left(0\right){\;R}_{\mathrm{yy}}\left(0\right)}},\;\mathrm{for}\;m=0\;\mathrm{with}\;R_{\mathrm{xy}}\left(m\right)=\{\begin{array}{c} \sum_{n=0}^{N-m-1}x_{n+m}{\;y}_{n},\;m\geq 0,\\ R_{\mathrm{xy}}\left(-m\right),\;m<0. \end{array}$$

A value ${\;\hat{R}}_{\mathrm{xy}}\left(0\right)$ near 0 or near 1 indicates a low or high similarity in temporal profiles of muscle activation between the affected and non-affected sides.

The CC coefficient is representative of the agonist–antagonist muscle couple coactivation. The stride sub-intervals above the threshold identifying on-off muscle status were employed to calculate the CC coefficient: overlapping sub-intervals for a duration higher than 30 ms between the couple of agonist-antagonist muscles contributed to measure the co-activation time as a percentage of the stride cycle [57].

The RMS coefficient is a representation of the activity of each muscle [58]. It was calculated on each normalized envelope signal x of N samples as follows:(2)$$\mathrm{RMS}=\;\sqrt{\frac{\sum_{n}x^{2}\left[n\right]}{N}}.$$

The sEMG outcomes were averaged from the two gait trials for each walking condition and were used for subsequent analyses.

Descriptive statistics were computed in order to appropriately explain the clinical and demographic characteristics of the sample. Data are represented for each recruited subject and as mean change scores (mean value ± standard deviation). The Friedman test was used to study differences among the three conditions (T1, T2, and o-RAGT_1_) for each variable. Successively, the Wilcoxon signed-rank test was used to study the differences between each condition (T1/T2 and T1/o-RAGT_1_). The statistical analysis was conducted at 95% CI, and a *p*-value < 0.05 two tailed was considered statistically significant. 

## 3. Results

Twenty-two subjects in the sub-acute phase post stroke were recruited in the study, and twenty of them trained with the o-RAGT, in addition to conventional therapy, without any adverse events (two subjects dropped out due to medical issues not associated to the training). As described in the previous paper of the authors [47], a number of barriers limited the implementation of the sEMG assessment due to subject-related, cultural barriers, technical, and administrative issues. Thus, as depicted in Figure 1, the sEMG was recorded during ecological overground gait at T1 and T2, and at o-RAGT_1_ in a sample composed of eight subjects (14 subjects agreed to participate in the instrumental assessment procedure but only 8 of them were able to walk for 10 m with assistance at T1). The demographic and clinical characteristics of each subject are depicted in Table 1. The statistical analysis of the clinical outcomes revealed significant improvement between T1 and T2 in MI-AL (*p*-value = 0.01) and FAC (*p*-value = 0.01). Specifically, the MI-AL and FAC showed a mean change score of 18.63 ± 9.35 and 1.88 ± 0.64, respectively. On the other hand, the MAS-AL (mean change score: −0.06 ± 0.68; *p*-value = 0.85), TCT (mean change score: 17.88 ± 13.79; *p*-value = 1), and 10MWT (mean change score: 0.22 ± 0.34 m/s; *p*-value = 0.11) did not registered significant differences.

The visual assessment of sEMG signals revealed heterogeneous volitional muscle activations at both T1 and T2, which did not consistently correlate with the typical activation timing of physiological gait. Technical barriers were encountered during o-RAGT_1_ [47]. Thus, the sEMG data of 3 subjects (S03, S11, S14) was not processed. Table 2 and Table 3 show the values of the BS coefficient and the CC coefficient, respectively. Data are depicted for each subject at T1 and T2 (during ecological overground gait), and at o-RAGT_1_ The BS coefficient increased between T1 and o-RAGT_1_ in a subset of subjects (S02 at TA and GM; S06 at TA, GM, BF, RF; S09 at TA, GM, BF; S11 at TA; and S18 at TA and BF). The effects of the 15 sessions of o-RAGT improved the BS of TA in subjects S02, S06, S09, S11, S20, of BF in S06, S20, and of RF in S06 only. The CC coefficient, on the other hand, did not show relevant decreases in distal muscles. Proximal muscles registered a decrease in CC between T1 and o-RAGT_1_ (S02, S06, S18, S20) and between T1 and T2 (S02, S11, S14, S18).

Figure 2 shows the RMS coefficients at T1, T2, and o-RAGT_1_ as mean and standard deviations values calculated on the whole sample. A mean increase of muscle activity (RMS) was registered at T2 in TA (affected and unaffected sides), GM (affected side), BF (affected and unaffected sides), and RF (unaffected sides). However, the standard deviations of RMS were high. The WPE (at o-RAGT_1_) increased the RMS coefficient of TA, GM, BF, and RF in the affected side only.

The Friedman test showed no-significant differences among the three conditions (T1, T2, o-RAGT_1_) in the following sEMG outcomes: BS-TA: *p*-value = 0.25; BS-GM: *p*-value = 0.45; BS-BF: *p*-value = 0.82; BS-RF: *p*-value = 1.20; CC-TA-GM_AL: *p*-value = 0.17; CC-BF-RF_AL: *p*-value = 0.55; CC-BF-RF_UL: *p*-value = 0.55; RMS-TA_AL: *p*-value = 0.17; RMS-GM_AL: *p*-value = 0.55; RMS-BF_AL: *p*-value = 1; RMS-RF_AL: *p*-value = 0.25; RMS-TA_UL: *p*-value = 0.82; RMS-GM_UL: *p*-value = 0.45; RMS-BF_UL: *p*-value = 0.54; RMS-RF_UL: *p*-value = 0.82. Only the CC coefficient measured between the TA and the GM of the unaffected limb revealed significant differences (*p*-value = 0.04), but the Wilcoxon test showed no-significant statistically differences. Table 4 and Table 5 depict the comparison of change scores of each sEMG outcome between T1 and T2, and T1 and o-RAGT_1_ respectively. Only the BS coefficient of the TA registered a significant difference (*p*-value = 0.04) between T1 and T2 with a large effect size (0.83).

## 4. Discussion

This study enrolled subjects subacute post stroke to assess the effects of o-RAGT and conventional therapy on muscle activation patterns. No adverse events were evidenced during the o-RAGT plus conventional training [14,15,16,35].

The sEMG data showed volitional muscle activation, although they were characterized by high variations in both amplitude and timing and did not correlate with the activation timing of physiological gait. The literature on gait recovery after stroke evidenced that sEMG patterns are heterogeneous [59] and do not change significantly over time in the subacute phase [60]. In this study, significant differences in sEMG outcomes were found in the bilateral symmetry of tibialis anterior between T1 and T2 only. This result is in accordance with the literature that found that the WPE promoted the neuromuscular coordination of lateral symmetry during gait [37]. However, it was not possible to find any correlation between the sEMG outcomes and the clinical results. The co-contraction (CC) seemed to decrease in proximal muscles at the end of the rehabilitation, thus suggesting improvements in proximal muscle activity. This outcome is in accordance with the study by Calabrò et al. [35] who analyzed the sEMG effects of o-RAGT with the same WPE. The sEMG changes could be related to the WPE functioning which actively supports the hip and knees and decreases the plantar flexion [34]. Since co-contraction could be correlated to spasticity, the o-RAGT seems not to increase spasticity in the subacute phase, consistent with other studies which employed robots for rehabilitation [14,15,16].

The analysis of sEMG during the first session of o-RAGT assessed the immediate effect of walking in a WPE in terms of muscle activation. No significant differences between the ecological overground gait at T1 and the oRAGT1 were found. This finding is in accordance with previous studies using a static exoskeleton where motor modules and activation signals were similar to overground locomotion [24,25]. Other studies on sEMG assessment during gait in a static exoskeleton evidenced a general lowering of muscle activity [26,31,33], maybe due to the uninvolving interaction between the subject and the robot. Conversely, our results showed that the WPE actively engaged the subjects, who showed volitional muscle activity, thus supporting the hypothesis that the overground WPEs stimulate the human-robot cognitive interaction [29,30,34]. This continuous active interaction between the subject and the WPE, in fact, directly stimulates the phenomena involved in the cognitive process and allows the subject to understand the potential of the robot, fully rely on the robot, and control the robot’s movements [8].

## 5. Limitations

The main limitations of the study were: the small sample size, the absence of follow-up assessments, and the heterogeneity of sEMG data. The small sample size may have been caused by a number of barriers to the implementation of sEMG assessment (subject compliance, setup procedure at o-RAGT_1_, as well as management and time-related issues), as described in the previous paper of the authors [47]. The absence of follow-up data was due to the subjects’ unavailability to come to the hospital for the assessments. The heterogeneity of sEMG data may have been caused by the procedures for subject enrollment. In the clinical study, the inclusion criteria did not take into account any restrictions on gait ability at baseline. The absence of a control group did not allow the study to isolate the effect of o-RAGT alone. However, the preliminary study showed positive effects of o-RAGT not only in subjects who could not walk but also in able walking subjects at baseline. This outcome is in accordance with clinical studies with the same WPE [11,16] that found that the WPE allows to conduct intensive overground gait training and move the lower limbs through pre-programmed trajectories, thus stimulating the neuromuscular system towards a physiological gait pattern. The hypothesis of the study was to determine if there was an improvement in muscle activation pattern, in terms of reduction of co-contractions and increased bilateral symmetry and activity. Unfortunately, the small sample size and the data heterogeneity limited our conclusions. In the future, the research agenda should include the implementation of novel procedures to promote the employment of electromyography in clinics.

## 6. Conclusions

The analysis of sEMG was conducted to assess the effects of o-RAGT and conventional therapy, in terms of muscle activation in subjects subacute post stroke. The study was conducted in a complex intensive clinical rehabilitation unit and found a set of barriers to the implementation of sEMG protocol. Nevertheless, the outcomes obtained in a small cohort of subjects revealed volitional muscle activations, a significant increase of bilateral symmetry in Tibialis Anterior muscles. Immediate positive effects of walking with a WPE were found in a subset of subjects. These findings suggest an active stimulation caused by the human-robot cognitive interaction. Future studies are necessary for understanding the neurophysiological effects of the combination of conventional therapy and overground robot-assisted gait training in subjects subacute post stroke. Moreover, the effective impact of the WPE on cognitive and psychological aspects should be ad-dressed.

## Figures and Tables

**Figure 1 brainsci-11-00448-f001:**
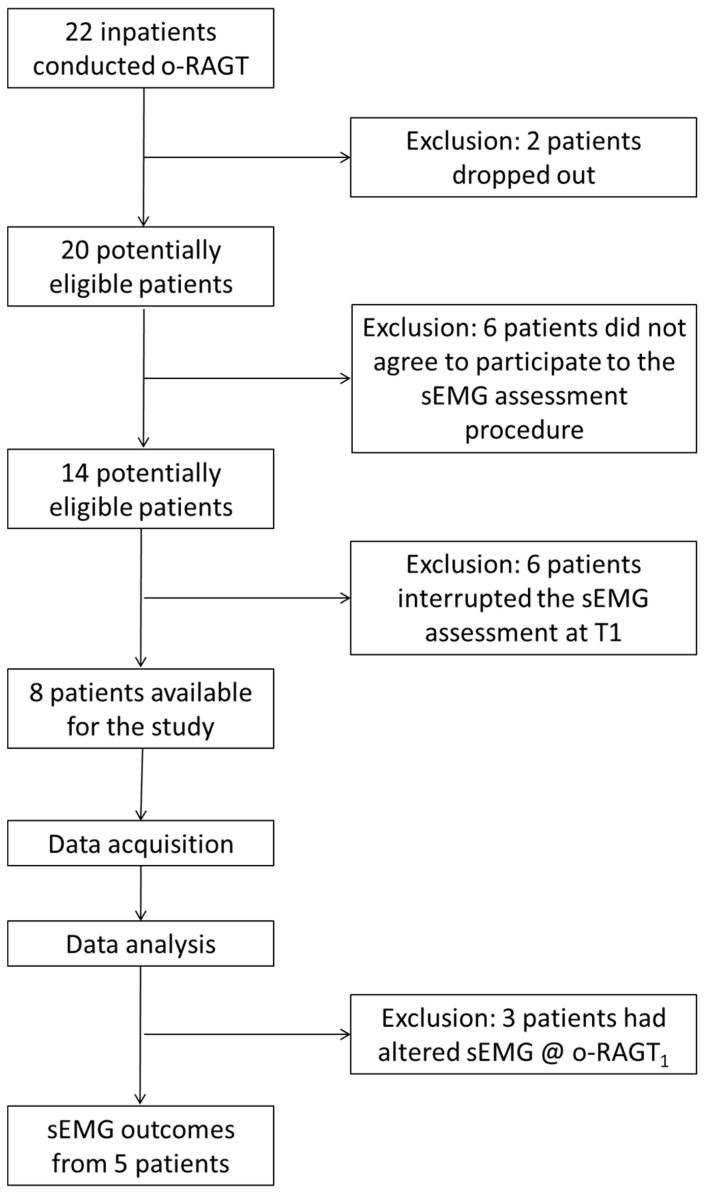
Flowchart of the study procedure.

**Figure 2 brainsci-11-00448-f002:**
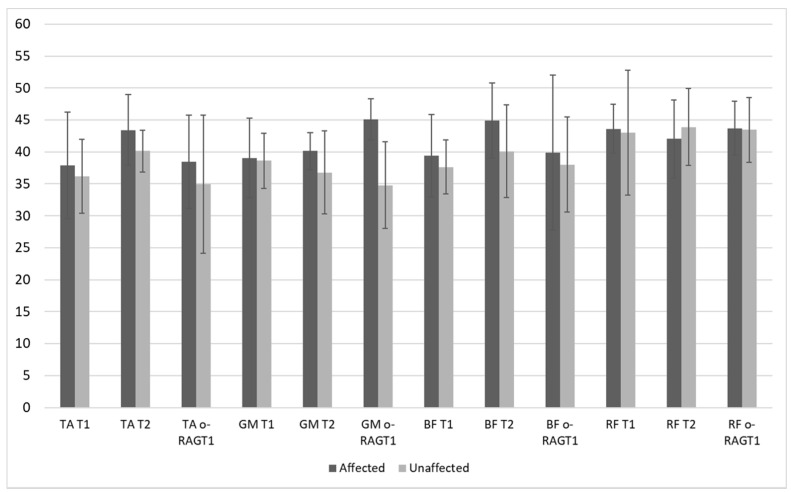
Root mean square (RMS) coefficient (mean values and standard deviations) at T1 and T2 during 10MWT (*N* = 8), and during the first session of o-RAGT, o-RAGT_1_ (*N* = 5).

**Table 1 brainsci-11-00448-t001:** Demographic and clinical characteristics of the sample.

ID	Age (Years)	Gender	AS	AOT (Days)	MAS-AL		MI-AL		FAC		TCT		10MWT (m/s) and Type of Assistive Device	
					T1	T2	T1	T2	T1	T2	T1	T2	T1	T2
S02	64	M	R	11	3.0	3.5	60	76	2	4	87	100	0.44 Wa	0.62 NoS
S03	69	M	L	13	0.0	0.0	64	76	1	3	74	100	0.34 Wa	0.48 TC
S06	54	F	L	29	1.0	0.0	48	65	1	3	61	100	0.16 AW	0.32 TC
S09	50	M	R	23	2.0	3.0	43	76	1	4	74	100	0.91 AW	0.67 NoS
S11	76	F	R	30	1.0	1.0	64	76	2	4	87	100	0.45 Wa	0.56 NoS
S14	44	F	R	26	2.0	2.0	53	76	2	4	74	100	0.38 Wa	0.53 NoS
S18	66	M	R	12	0.0	0.0	76	82	4	5	100	100	0.44 NoS	1.41 No
S20	66	M	R	75	2.0	1.0	70	100	3	4	100	100	1.09 TC	1.38 NoS

Abbreviations: Male (M); Female (F); Affected Side (AS); Acute event Onset Time (AOT); Right (R); Left (L); Modified Ashworth Scale Affected lower Limb (MAS-AL); Motricity Index Affected lower Limb (MI-AL); Functional Ambulation Classification (FAC); Trunk Control Test (TCT); 10-Meter Walking Test (10MWT).Type of assistive device: Axillary Walker (AW); Walker (Wa); Tripod Cane (TC); None with Supervision (NoS); None (No).

**Table 2 brainsci-11-00448-t002:** Bilateral symmetry (BS) coefficient at T1 and T2 during 10MWT, and during the first session of overground Robot-Assisted Gait Training (o-RAGT) (o-RAGT_1_).

ID	TA			GM			BF			RF		
	T1	o-RAGT_1_	T2	T1	o-RAGT_1_	T2	T1	o-RAGT_1_	T2	T1	o-RAGT_1_	T2
S02	0.64	0.83	0.71	0.59	0.77	0.56	0.85	0.84	0.82	0.83	0.84	0.71
S03	0.83	n.a.	0.80	0.75	n.a.	0.46	0.77	n.a.	0.82	0.87	n.a.	0.80
S06	0.70	0.62	0.78	0.59	0.91	0.64	0.50	0.83	0.86	0.63	0.87	0.78
S09	0.65	0.90	0.89	0.82	0.88	0.63	0.85	0.93	0.82	0.92	0.90	0.89
S11	0.68	n.a.	0.71	0.68	n.a.	0.80	0.70	n.a.	0.68	0.85	n.a.	0.71
S14	0.73	n.a.	0.73	0.79	n.a.	0.75	0.81	n.a.	0.83	0.86	n.a.	0.73
S18	0.81	0.87	0.67	0.91	0.84	0.61	0.72	0.85	0.67	0.84	0.80	0.67
S20	0.81	0.80	0.87	0.83	0.82	0.84	0.80	0.68	0.84	0.77	0.77	0.87

Abbreviations: Tibialis Anterior (TA); Gastrocnemius Medialis (GM); Biceps Femoris (BF); Rectus Femoris (RF); not available (n.a.).

**Table 3 brainsci-11-00448-t003:** Co-contraction (CC) coefficient at T1 and T2 during 10MWT, and during the first session of o-RAGT (o-RAGT_1_).

ID	TA-GM_AL			TA-GM_UL			BF-RF_AL			BF-RF_UL		
	T1	o-RAGT_1_	T2	T1	o-RAGT_1_	T2	T1	o-RAGT_1_	T2	T1	o-RAGT_1_	T2
S02	12.51	12.10	20.52	18.44	22.39	38.09	32.51	15.54	22.85	43.81	33.26	36.42
S03	24.40	n.a.	11.90	26.01	n.a.	10.02	21.45	n.a.	29.71	23.30	n.a.	33.56
S06	17.95	11.16	34.60	11.01	7.39	11.02	25.48	14.38	30.01	10.78	32.74	38.64
S09	23.40	26.80	33.73	22.45	26.46	27.13	15.42	45.62	27.78	23.57	32.64	38.47
S11	11.71	n.a.	31.30	37.81	n.a.	30.33	31.27	n.a.	29.05	41.36	n.a.	35.27
S14	27.33	n.a.	35.96	24.36	n.a.	35.88	40.03	n.a.	52.33	43.86	n.a.	36.58
S18	18.58	25.64	22.82	10.01	22.40	21.28	36.39	32.88	29.43	30.82	49.11	23.20
S20	15.76	25.42	25.59	24.51	28.40	39.99	22.06	17.48	31.68	26.67	18.22	28.90

Abbreviations: Tibialis Anterior-Gastrocnemius Medialis Affected lower Limb (TA-GM_AL); Tibialis Anterior-Gastrocnemius Medialis Unaffected lower Limb (TA-GM_UL); Biceps Femoris-Rectus Femoris Affected lower Limb (BF-RF_AL); Biceps Femoris-Rectus Femoris Unaffected lower Limb (BF-RF_UL); not available (n.a.).

**Table 4 brainsci-11-00448-t004:** Comparison of T1 and T2 change scores of each surface ElectroMyoGraphy (sEMG) outcome.

*N* = 8		Mean Change Score ± SD			Wilcoxon Test	
					Effect Size	*p*-Value
BS						
	TA	0.04	±	0.11	0.83	0.04
	GM	−0.08	±	0.16	−0.39	0.38
	BF	0.04	±	0.13	0.33	0.46
	RF	−0.05	±	0.12	−0.22	0.64
CC						
	TA-GM_AL	8.10	±	9.66	0.50	0.25
	TA-GM_UL	4.89	±	12.12	0.39	0.38
	BF-RF_AL	3.53	±	8.72	0.33	0.46
	BF-RF_UL	3.36	±	13.21	0.11	0.84
RMS						
	TA_AL	0.04	±	0.12	0.39	0.38
	GM_AL	−0.01	±	0.12	0.00	1.00
	BF_AL	0.05	±	0.12	0.44	0.31
	RF_AL	−0.02	±	0.09	−0.33	0.46
	TA_UL	0.02	±	0.06	0.44	0.31
	GM_UL	−0.03	±	0.05	−0.50	0.25
	BF_UL	0.01	±	0.13	0.06	0.95
	RF_UL	0.00	±	0.08	−0.11	0.84

Abbreviations: Tibialis Anterior (TA); Gastrocnemius Medialis (GM); Biceps Femoris (BF); Rectus Femoris (RF); Tibialis Anterior-Gastrocnemius Medialis Affected lower Limb (TA-GM_AL); Tibialis Anterior-Gastrocnemius Medialis Unaffected lower Limb (TA-GM_UL); Biceps Femoris-Rectus Femoris Affected lower Limb (BF-RF_AL); Biceps Femoris-Rectus Femoris Unaffected lower Limb (BF-RF_UL).

**Table 5 brainsci-11-00448-t005:** Comparison of T1 and o-RAGT_1_ change scores of each sEMG outcome.

*N* = 5		Mean Change Score ± SD			Wilcoxon Test	
					Effect Size	*p*-Value
BS						
	TA	0.08	±	0.14	0.47	0.44
	GM	0.10	±	0.16	0.47	0.44
	BF	0.08	±	0.17	0.47	0.44
	RF	0.04	±	0.11	−0.07	1.00
CC						
	TA-GM_AL	2.58	±	6.47	0.73	0.19
	TA-GM_UL	4.12	±	5.67	0.87	0.13
	BF-RF_AL	−1.19	±	18.37	−0.33	0.63
	BF-RF_UL	6.06	±	14.98	0.47	0.44
RMS						
	TA_AL	0.03	±	0.04	0.73	0.19
	GM_AL	0.05	±	0.09	0.20	0.81
	BF_AL	0.01	±	0.13	−0.07	1.00
	RF_AL	0.00	±	0.04	−0.07	1.00
	TA_UL	0.02	±	0.13	0.07	1.00
	GM_UL	−0.05	±	0.06	−0.87	0.13
	BF_UL	0.02	±	0.07	0.47	0.44
	RF_UL	0.02	±	0.13	−0.07	1.00

Abbreviations: Tibialis Anterior (TA); Gastrocnemius Medialis (GM); Biceps Femoris (BF); Rectus Femoris (RF); Tibialis Anterior-Gastrocnemius Medialis Affected lower Limb (TA-GM_AL); Tibialis Anterior-Gastrocnemius Medialis Unaffected lower Limb (TA-GM_UL); Biceps Femoris-Rectus Femoris Affected lower Limb (BF-RF_AL); Biceps Femoris-Rectus Femoris Unaffected lower Limb (BF-RF_UL).

## Data Availability

The data presented in this study are available on request from the corresponding author.

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
