# Peer review of "Functional Gait Recovery after a Combination of Conventional Therapy and Overground Robot-Assisted Gait Training Is Not Associated with Significant Changes in Muscle Activation Pattern: An EMG Preliminary Study on Subjects Subacute Post Stroke"

_brainsci, 2021, doi:10.3390/brainsci11040448_

Round 1

Reviewer 1 Report

This study represents a secondary analysis of the data collected in a larger RCT (NCT03395717) that recruited a subacute stroke sample undergoing inpatient rehabilitation to examine the effects of overground robot-assisted gait training (o-RAGT) provided as an add-on to the standard inpatient therapy, which served as a control. Several main outcomes of this RCT have already been reported whereas this study is comparing EMG findings in the major lower limb muscles from before (T1) to after the end of treatment (T2) in a subsample of 8 participants assigned to the o-RAGT group (data also reported for 4 subjects while using o-RAGT for the first time).

In general, this type of study is viewed favorably since it is still unknown how gait training in o-RGAT affects muscle activation patterns in the affected and non-affected legs. Also, the data collection and analysis procedures seem to be executed according to the expected standards. The manuscript is for the most part well written and the illustrations are informative.  However, several major concerns deserve more attention. Minor comments are also provided in an attempt to improve text clarity.

  1. Although the authors acknowledge the difficulty with collecting EMG data, there is no explanation as to why the study did not include any control participants. This is problematic because the comparison of EMG between the two groups was stated as a secondary goal in the trial registry. Therefore, the authors are encouraged to incorporate the control data into the report but if unable to do so, a clear explanation needs to be provided to avoid the perception of selective reporting, which is one of the main reasons for the existence of a trial registry.
  2. Short of comparison between the experimental and control groups, the title and the entire text (particularly the aim statement at the end of the Introduction) should not state the objective and ascribe the reported effects to an o-RAGT alone but to the combination of o-RAGT and standard inpatient therapy since the distinct effect of the o-RAGT cannot be isolated with the present study design.

Specific Comments

Introduction:

  1. in the statements referring to the report by Androwis (ref 34), the sample should be better characterized in terms of the size, time since stroke, and motor/gait abilities (overground speed, gait outcomes, etc) so the reader can understand the similarities and differences between this and previous report in stroke subjects.
  2. The research question is not obvious, please state it more clearly and specifically.

Methods

  1. It is not clear how this can be considered a pilot study when it was stated as a secondary objective in the trial registry.
  2. State if data for this report were collected at a single center or more than one center.
  3. It should be clearly stated why week 8 and 20 follow-up data were not included in this report.
  4. It is not clear if the o-RAGT1 gait evaluation occurred on the same day as the baseline evaluation.
  5. Outcome measures: “screened” is not the best term here, consider “assessed” instead.
  6. The trial/task terminology is confusing and not as typically used: what is now labeled as the “trial” should be called either a task or better yet a condition (overground, o-RAGT1), whereas the “task” should be termed a “trial” (e.g., two trials were collected for each condition). Note that the term task should be replaced with a trial in the last paragraph before the Results section.
  7. The statement in which Rxy values (0, 1) were interpreted is a bit confusing. Consider “indicates a low or high similarity in temporal profiles of muscle activation between the affected and non-affected sides”.

Results

  1. Figure 1 should also include a reference to 4 subjects in which o-RAGT1 evaluation was done (Exclusion, similar to where it is said that 6 cases this not walk at T1).
  2. Indicated in Table 1/Column 10MWT which subjects used what assistive device during the T1 and T2 assessment sessions.
  3. Although the Methods section correctly refers to subjects, the Results section switches to patients, which is not appropriate- choose subjects or participants and accordingly changes the labels in all tables and the text from PAT to either S or P.
  4. It is said that EMG data were not analyzed for 4 subjects (ID 03, 11, 14, 19), which would imply that it was done in 4 subjects. Yet the tables of results never refer to ID19 and include EMG data for 5 subjects. This is confusing, please clarify; perhaps revising Figure 1 can also help in this regard.
  5. Figure 2 caption: make a note that o-RAGT1 data are from 5 subjects.

Discussion/Conclusion

  1. Make sure the Discussion and Conclusion are correctly ascribing the effect to the combination of standard therapy and o-GART training.
  2. In the Limitations section, replace “comprehend” with “account for”; indicate that the effect of o-RAGT could not be isolated in this study due to no control data; address the limitation of the results regarding questionable relevance of using o-RAGT in people who are already ambulatory within a month after stroke.

Author Response

Please find attached the Reply to the Review Reports. Best regards

Reviewer 2 Report

Process of recovery would be varied according to the day from stroke onset, stroke severity, lesion site, or patients’ age. It is difficult to lead a conclusion of the present study because of a small sample size and little restrictions of the entry criteria. Description of the result is subjective, and reviewer cannot understand the different among “notable”, “moderately”, and “little” improvement. Reviewer cannot find the data of Patient 19, whose sEMG data was not prcessed.

Author Response

(The authors gave the same response as above.)

Reviewer 3 Report

The benefits of robotic gait training is of interest to patients, physicians and rehabilitation clinicians.  It is of particular interest for patients post stroke and those with spinal cord injury.  Some research suggests robotic technology can maximize plasticity and neural recovery.  This type of technologically assisted training is known to be expensive but also can be life changing to the patient. This technology is sophisticated and often presents technical problems with training. 

This paper is very difficult to read, particularly to understand the EMG changes they hypothesized to find. The authors need to make sure the reader can follow the importance of the data findings ( e.g. symmetry and contraction coefficients for the selected muscles of the lower limb and potential similarities in muscle firing patterns for the affected and unaffected muscle pairings) .  Also, the authors need to refer to patients sub acute post stroke.  Each subject is an individual with a stroke. The way it reads is the subject is a stroke.  This change is necessary in the title as well as the text. 

The tables are generally hard to read and maybe vertical lines between the different variables would make the data easier to read.  Although there is a lot of EMG data collected, the number of subjects was too small to justify parametric statistics.  Perhaps they should provide mean change scores in all of the data and calculate effect sizes.  This might provide more insightful findings than the ANOVA data.

In Table 1 the authors provide information  on change  in  dependent variables relative to gait from T1-T2 .  If effect sizes and/or mean change scores were calculated, it looks like there was improvement in gait ambulation speed (which appears to meet minimally clinically significant gains). Similarly it  looks like there could have been gains in Mortricity of the affected lower limb, functional ambulation and trunk control without noticeable  improvement in the Ashworth Scores. Although it was not the purpose of the study to report this information, it would allow the researchers to report that these functional gains were not correlated with significant changes in muscle firing patterns. 

The researchers tried to pull some individual subject data from the group analysis.  This can be in the discussion but not included in the  results or conclusions.

There is a section on study limitations.  Obviously the big problem is only 8 subjects.  There are significant differences in the impact of a stroke on individual patients.  Thus, one either needs to have a lot more patients to find a trend.  However more patients will not likely show normal data distribution in this population of patients.  It is possible one could find some trends if the analysis was changed to nonparametric analysis.  Figure 2 provides some trend information .  The Root Mean Square coefficient was higher on the aftected leg compared to the unaffected leg  for TA GM and BF.  In addition, for the subjects and the controls, T2 was higher than T1. for both subjects and control for TA, BF and  unaffected for RF.  What does that mean?  Was that expected?  

The title is ok but maybe  would be more informative with something like:

 In Patients Subacute Post Stroke, Functional Gains in Gait Post Overground Robot-Assisted Gait training is Not Associated with Significant Changes in Muscle EMG Firing Patterns

Author Response

(The authors gave the same response as above.)

Round 2

Reviewer 1 Report

The manuscript has been improved through revision. 

Author Response

Thank you for your comment. Best regards

Reviewer 2 Report

Revised version is interesting and improved in scintic soundness.

Author Response

(The authors gave the same response as above.)

Reviewer 3 Report

The authors did extensive editing and the manuscript is improved.  The small number of subjects is still  a major issue.  Is it possible to get 2-3 more subjects?  If you added a few subjects and reanalyzed the data it would be interesting to see if your findings are the same.  It would make a resubmission better.   However, additional editing is still needed on the manuscript with the current number of subjects. (see attachment)

Author Response

Thank you for your comment.

In attachment the answers to your comments.
Best regards
